# Numerical Studies of the Effects of the Substrate Structure on the Residual Stress in Laser Directed Energy Additive Manufacturing of Thin-Walled Products

**Hang Jing [1], Peng Ge [1,2,*], Zhao Zhang [3], Jun-Qi Chen [1], Zhong-Ming Liu [2] and Wei-Wei Liu [4]**

1 College of Civil Engineering and Architecture, Henan University of Technology, Zhengzhou 450000, China; hangjing@haut.edu.cn (H.J.); chenjunqi@haut.edu.cn (J.-Q.C.)

2 Zhengzhou Research Institute of Mechanical Engineering Co., Ltd., Zhengzhou 450052, China; iuzm@zrime.com.cn

3 State Key Laboratory of Structural Analysis for Industrial Equipment, Department of Engineering Mechanics, Faculty of Vehicle Engineering and Mechanics, Dalian University of Technology, Dalian 116024, China; zhangz@dlut.edu.cn

4 School of Mechanical Engineering, Dalian University of Technology, Dalian 116024, China; liuww@dlut.edu.cn

* Correspondence: gp101022@haut.edu.cn

**Abstract:** A new method of controlling the residual stress in laser directed energy deposition additive manufacturing (DED AM) products proposed based on constraints used in manufacturing and the substrate design. The simulation results of the residual stress, which were validated with the experimental measured data, showed that weaker constraints on the substrate could greatly decrease the residual stress in the laser DED AM products. In addition, by designing local reduced thickness regions into the substrate, such as long strip holes or support legs, the residual stress in DED AM products could be further decreased. In this study, when long strip holes were designed in the substrate, the tensile residual stress was decreased by 28%. An even smaller amount of residual stress was achieved when the design structure was changed to support legs. The tensile residual stress decreased by more than 30%. The fewer support legs, the smaller the residual stress. The residual stress in DED AM products could be well-controlled by design, while the stiffness can be weakened with fewer constraints.

**Keywords:** residual stress; thermal distortion; substrate structure; directed energy deposition

## 1. Introduction

In recent years, laser directed energy deposition additive manufacturing (DED AM), an advanced technology that builds materials layer by layer, has developed quickly and gained increasing attention [1–3]. It has been widely used in the manufacturing of critical products in aerospace [4] and bioengineering [5]. However, the residual stresses formed in the multiple thermal processes, similar with that in the welding [6,7], still significantly influence the mechanical properties of the AM products, especially their bearing capacity and fatigue life [8,9].

In DED AM, the heat energy density absorbed by the deposition layer is the key factor in the local temperature gradient and in inducing high residual stress. It can be greatly influenced by the deposited materials, laser power, laser scanning speed, laser beam radius, and structure of the substrate. In the past years, the effect of the DED AM parameters on the residual stress has been given more attention [10–16]. Ha et al. [10] studied the effects of the temperature and progressive solidification on residual stress and showed that distortion of the solid-liquid interface led to a decrease in the residual stress. Mukherjee et al. [11] found that the residual stress in the laser additive manufacturing products can be greatly decreased by reducing the layer thickness. Zhan et al. [12] reported that the residual

stress increased with the increase in laser power but decreased with the increase in the laser scanning speed and the powder feed rate. In addition, the effects of the laser scanning path strategy [13,14], the laser-particle interaction [15], and the deposited materials [16] on the residual stress and residual distortion in DED AM products have all attracted more attention. Meanwhile, a lot of methods for controlling the residual stress in DED AM products were also proposed [17–19] based on achieved findings in welding [20,21]. As a few examples, Nie et al. [17] stated that the application of a static magnetic field in the DED AM could greatly reduce the residual stress in a deposited Inconel 718 product. Zhan et al. [18] and Santo-aho et al. [19] studied the influence of the post-heat-treatment on residual stress. The results showed that the high tensile residual stresses in the deposition layers could be clearly decreased and even changed into compressive stress after the heat treatment. During the heat treatment, the cooling rate and solution temperature, when compared with the aging temperature and aging time, are more important factors in reducing the residual stress.

Though the influencing factors and the method of controlling the residual stress in DED AM products have been systematically studied, less attention has been given to research on the effect of the substrate structure on residual stress. Based on the research on residual stress [22], the inhomogeneous thermal deformation in the heating and cooling process is the key factor for the formation of the residual stress. The high manufacturing temperature leads to the yield of the deposited materials. As a result, the plastic is strained and cannot be released before the removal of the constraints. The stiffness and the geometric configuration can greatly influence the expansion and shrinkage of the deposited layers in DED AM [23–25]. It is necessary to put forward a new study on the effect of the substrate structure and the constraints on residual stress. In this paper, a sequentially coupled thermomechanical model [26,27] was utilized to simulate the residual stresses in DED AM products with different substrate structures and constraints. Then an experiment was also developed to validate the simulation results. Finally, the method for controlling the residual stress in DED AM products was presented by designing the substrate structure.

## 2. Experiment and Materials

In the experiment, Ti-6Al-4V powder particles were deposited on a substrate with dimensions of $150 \times 20 \times 10$ mm using the laser deposition additive manufacturing (DED AM) system shown in Figure 1a. In the DED AM system, the FCL-2000 Semiconductor Laser, which is manufactred by the Changchun laser technology Co., Ltd, Changchun, China, with a maximum power of 2 kW and a wavelength in the range of 900–1000 nm, the AK190 additive manufacturing head, and the powder feeder were all made in China. Ten deposition layers with dimensions of $100 \times 50 \times 0.3$ mm were built on the substrate by providing Ti-6Al-4V particles from the tube cell. The manufacturing process is shown in Figure 2. The laser was moved from the initial manufacturing point to the other side with a designed laser moving velocity. Thus, the Ti-6Al-4V particles could be melted in the melt pool on the substrate or the previous layer. Constraints on one end and on the bottom surface of the substrate were respectively applied in the manufacturing process of the DED AM product. Meanwhile, an infrared thermal imager FLIR T430s (Philip, Wilsonville, OR, USA) was used in the experiment to measure the manufacturing temperature of the deposited sample. After production, the residual distortion of the deposited sample was measured using the three-coordinate measuring machine shown in Figure 1b. The deposited sample was fixed by precision flat-nosed pliers and laid on the clamping table. A ruby probe was also used in the experimental process. In the experiment, various measurement points were automatically selected by the three-coordinate measuring machine with the movement of the ruby probe. Thus, the spatial coordinates could be sequentially obtained. By dealing with the data of the automatically selected points on the measured line or measured face based on the coordinate origin set in the three-coordinate measuring machine, the residual stress of the laser DED AM product could be obtained.

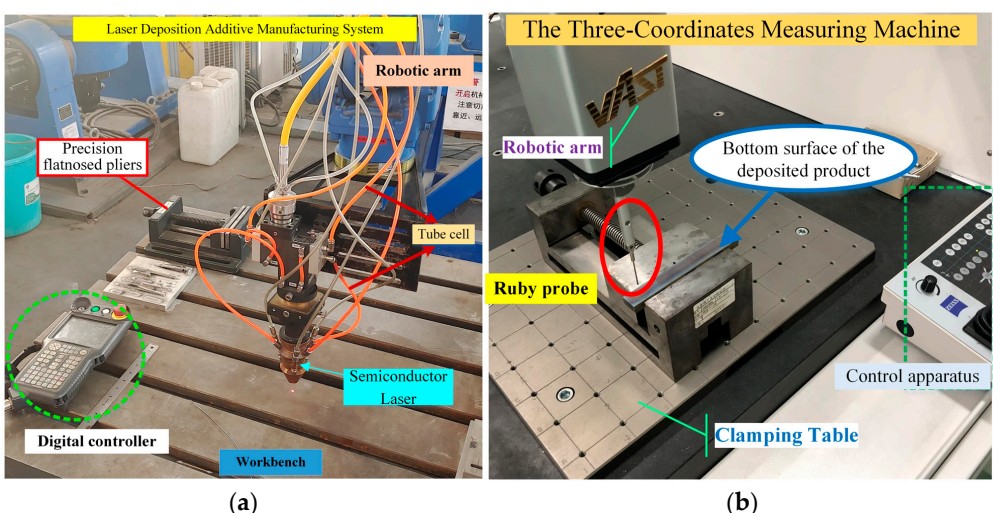

**Figure 1.** Laser directed energy deposition additive manufacturing and the three-coordinate measuring machine: (**a**) laser deposition additive manufacturing system; (**b**) the three-coordinates measuring machine.

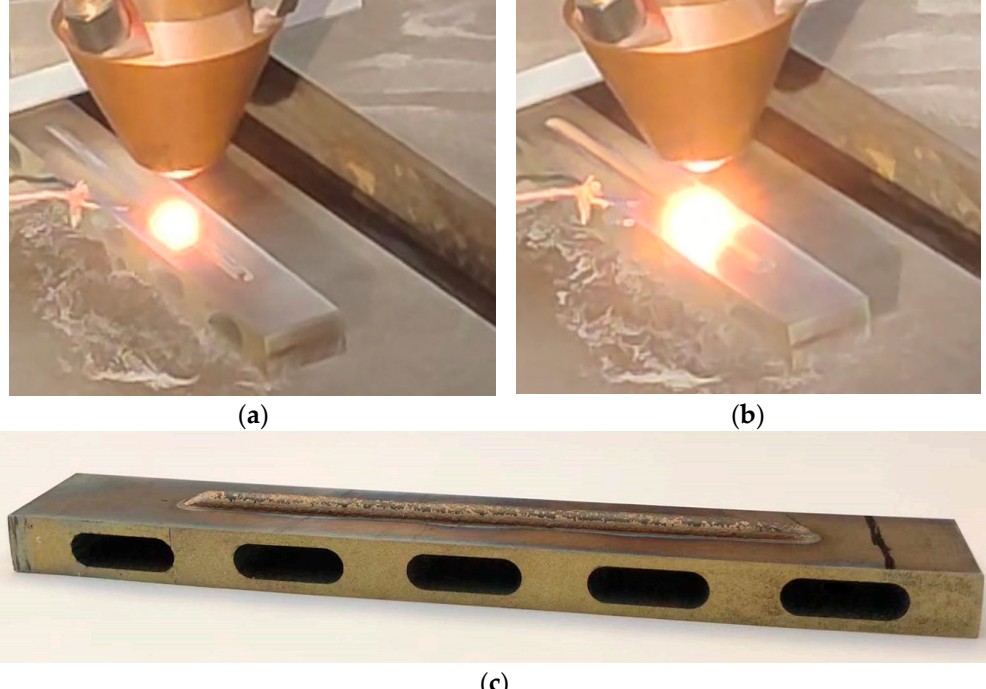

**Figure 2.** The laser DED AM process: (**a**) at 38 s; (**b**) at 175 s; (**c**) the laser DED AM product.

The obtained experimental data about the temperature and the residual distortion were then used in the validation of the numerical simulation. The chemical composition of the Ti-6Al-4V particles is listed in Table 1 [28] and the manufacturing parameters in the experiment are listed in Table 2.

**Table 1.** Chemical composition of the Ti-6Al-4V particles [28].

| Elements | Ti | Al | V | Fe | C | O | N | H |
|---|---|---|---|---|---|---|---|---|
| wt.% | Balance | 6.30 | 4.18 | 0.028 | 0.008 | 0.096 | 0.003 | 0.001 |

**Table 2.** Manufacturing parameters used in the experiment.

| Manufacturing Parameter | Value |
| --- | --- |
| Laser power, $Q$/(W) | 600 |
| Scanning speed, $v_l$/(mm/s) | 5 |
| Laser beam diameter, $d$/(mm) | 5 |
| Deposition length, $l_d$/(mm) | 100 |
| Deposition height, $h_d$/(mm) | 3 |
| Lifting capacity along the vertical direction, $h_z$/(mm) | 0.3 |

## 3. Numerical Model

### 3.1. Thermal Analysis

In this paper, using the same manufacturing parameters as in the experiment, a sequentially coupled thermomechanical model [29,30] was established using ABAQUS. First, the manufacturing temperature of the deposited sample was modeled in the heat transfer simulation using the moving heat source. The temperature results were then used as the initial temperature conditions in the mechanical modeling, which had the same element mesh, but a different element type. Based on the studies of temperature distribution by Michaleris [31] and residual stress by Stender [32], there was no gas pore or defect in the powder particles that were used. There was no difference between the thermal properties of the Ti-6Al-4V particles and the Ti-6Al-4V substrate. When the particles reached the substrate or the deposition layer, they were melted in the melt pool. The impurities or oxygen atoms were all blocked outside the melt pool by the insertion of shielding gas used in laser DED AM. So, the temperature-dependent thermal properties of the solid Ti-6Al-4V, which are shown in Figure 3, were used in the modeling of the heat transfer process. Though the melting temperature of Ti-6Al-4V is 1660 °C, the temperature-dependent thermal mechanical properties of Ti-6Al-4V above 1200 °C up to 1660 °C was calculated using the backward difference method. In particular, in the heat transfer modeling, the latent heat of 365 kJ/kg of Ti-6Al-4V when the solidus temperature is 1600 °C and the liquid temperature is 1660 °C was also considered based on [33]. Meanwhile, birth and death elements were used in the simulation for modeling the sequential deposition of the Ti-6Al-4V. The elements were activated by moving the laser heat source.

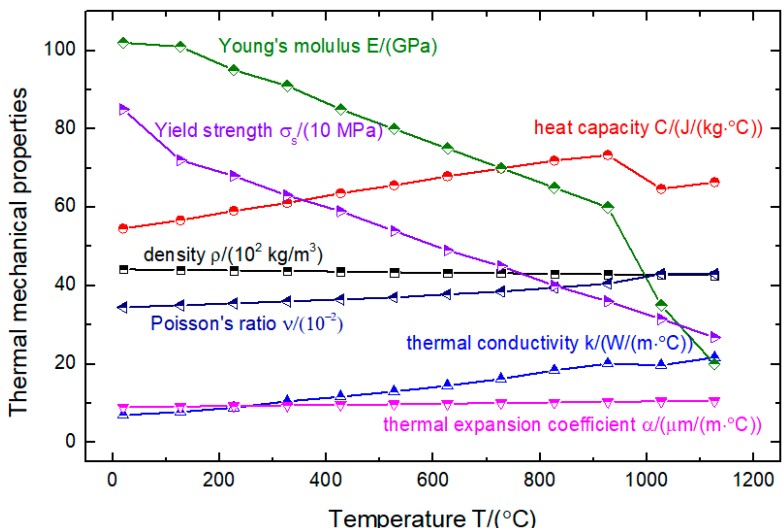

**Figure 3.** Temperature-dependent thermal mechanical properties of Ti-6Al-4V.

In the heat transfer simulation, the temperature field, $T$, needed to satisfy the energy balance equation, which can be written as [34]:

$$\rho C \frac{\partial T}{\partial t} = \nabla \cdot (k \nabla T) + q(x, y, z, t), \tag{1}$$

where $t$ is time, $k$ is the thermal conductivity coefficient, $\rho$ is density, $C$ is specific heat, and $q(x, y, z, t)$ is the volumetric heat flux. In deep penetration welding, the double ellipsoid heat source, which was first developed by Goldak [35], was widely used to better model the distribution of the heat energy in electron beam and laser welding. The heat source describes the changing of the heat energy with the moving of the electron beam or laser more clearly. In the numerical modeling of the manufacturing temperature in laser DED AM, Yang et al. [1], Michaleris [31], and Heigel et al. [36] all used the double ellipsoid heat source to model the laser heat. By using a double ellipsoid heat source model, the volumetric heat flux, $q(x, y, z, t)$, can be written as

$$q_f(x, y, z, t) = \frac{6\sqrt{3}\eta Q f_f}{a_f b c \pi \sqrt{\pi}} \exp\left\{ -3\left[ \left(\frac{x + v_l t}{a_f}\right)^2 + \left(\frac{y}{b}\right)^2 + \left(\frac{z}{c}\right)^2 \right] \right\}, \tag{2}$$

for the front part of the double ellipsoid and

$$q_r(x, y, z, t) = \frac{6\sqrt{3}\eta Q f_r}{a_r b c \pi \sqrt{\pi}} \exp\left\{ -3\left[ \left(\frac{x + v_l t}{a_r}\right)^2 + \left(\frac{y}{b}\right)^2 + \left(\frac{z}{c}\right)^2 \right] \right\}, \tag{3}$$

for the rear part of the double ellipsoid, where $a_f$, $a_r$, $b$, and $c$ are, respectively, the axes of the $X$, $Y$, and $Z$ directions, $\eta$ is the coefficient of the absorption of the laser power by the materials in the deposition layers, $Q$ is the laser power, $f_f$ and $f_r$ are the heat input coefficients of the two semi-ellipsoids, and $v_l$ is the laser scanning speed. The length, width, and depth of the double ellipsoid heat source model were selected based on [37].

To solve the governing energy balance equation of the manufacturing temperature, $T$, the thermal convection and radiation boundary conditions are considered and are written as Equations (4) and (5) [38]:

$$q_{con} = -h(T - T_a), \tag{4}$$

$$q_{rad} = -\varepsilon_{rad}\sigma_{rad}\left(T^4 - T_a^4\right), \tag{5}$$

where $h$ is the convective coefficient, $T_a$ is the ambient temperature, $\varepsilon_{rad}$ is the emissivity of the deposition layer, and $\sigma_{rad}$ is the Stefan-Boltzmann constant for radiation. The parameters used in the heat transfer model are listed in Table 3.

**Table 3.** Parameters used in the heat transfer simulation.

| Numerical Parameters | Value |
|---|---|
| positive direction of $X$ axis, $a_f$/(mm) | 2.5 |
| negative direction of $X$ axis, $a_r$/(mm) | 5 |
| positive direction of $Y$ axis, $b$/(mm) | 2.5 |
| positive direction of $Z$ axis, $c$/(mm) | 0.3 |
| heat input coefficients of the front part of the double ellipsoid, $f_f$ | 0.7 |
| heat input coefficients of the rear part of the double ellipsoid, $f_r$ | 1.3 |
| absorption coefficient of laser power, $\eta$ | 0.7 |
| convective coefficient, $h$/W/(m$^2$/°C) | 30 |
| ambient temperature, $T_a$/°C | 20 |
| emissivity of deposition layer, $\varepsilon_{rad}$ | 0.2 |
| Stefan-Boltzmann constant for radiation, $\sigma_{rad}$ | $5.6704 \times 10^{-8}$ |

### 3.2. Mechanical Analysis

In the mechanical simulation, the governing equation for the thermal stress can be written as:

$$\mathrm{div}\,\boldsymbol{\sigma} = 0, \tag{6}$$

where $\boldsymbol{\sigma}$ is the stress tensor. Based on the low hardening behavior of Ti-6Al-4V, the elastic/perfectly plastic constitutive law is expressed as:

$$\boldsymbol{\sigma} = \mathbf{C} : \boldsymbol{\varepsilon}_e, \tag{7}$$

$$\boldsymbol{\varepsilon}_e = \boldsymbol{\varepsilon} - \boldsymbol{\varepsilon}_p - \boldsymbol{\varepsilon}_T, \tag{8}$$

where $\mathbf{C}$ is the stiffness tensor, $\boldsymbol{\varepsilon}$ is the total strain tensor, $\boldsymbol{\varepsilon}_p$ is the plastic strain tensor, and $\boldsymbol{\varepsilon}_T$ is the temperature strain tensor, which can be written as:

$$\boldsymbol{\varepsilon}_T = \alpha(T - T_{\mathrm{initial}}) \cdot \mathbf{I}, \tag{9}$$

where $\alpha$ is the thermal expansion coefficient, $T_{\mathrm{initial}}$ is the initial temperature of the deposited materials, and $\mathbf{I}$ is the identity matrix. The inhomogeneous temperature strain caused in the expansion and shrinkage process is the main factor of plastic strain and thermal stress. The higher the manufacturing temperature, $T$, and the larger the thermal expansion coefficient, $\alpha$, the larger the plastic strain. As a result, when the constraints are removed, more plastic strain cannot be relaxed, and then greater residual stress is found in the deposition structure.

In this paper, the effect of constraints on the substrate in the heating process in DED AM on residual stress was first studied with the numerical models shown in Figure 4a,b. Based on the different fixture forms in the experiment, the fixed constraints on the bottom surface (constraint 1) and the one end (constraint 2) of the substrate were respectively used as the displacement boundary conditions in the mechanical analysis. Then, the different substrate structures, the substrates designed with long strip holes and with support legs, were built in ABAQUS, as shown in Figure 5a,b, respectively. The effect of the substrate structure on the residual stress in the thin-walled deposited products manufactured by DED AM was further studied.

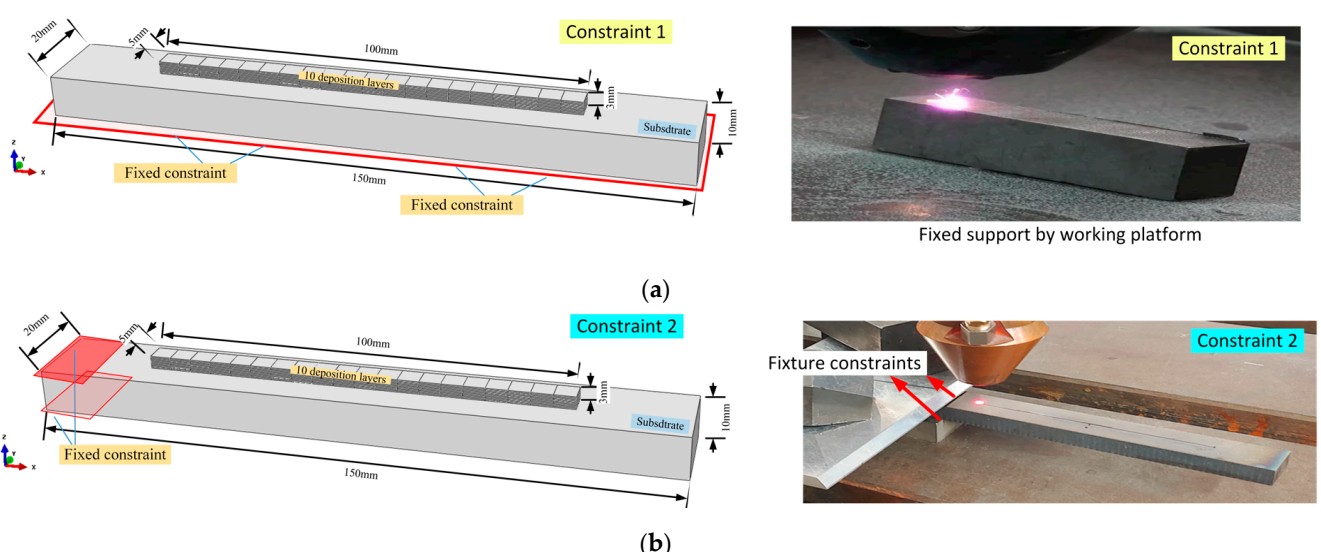

**Figure 4.** Different constraints on the substrate used in DED AM: (**a**) constraints on bottom surface; (**b**) constraints on the one end.

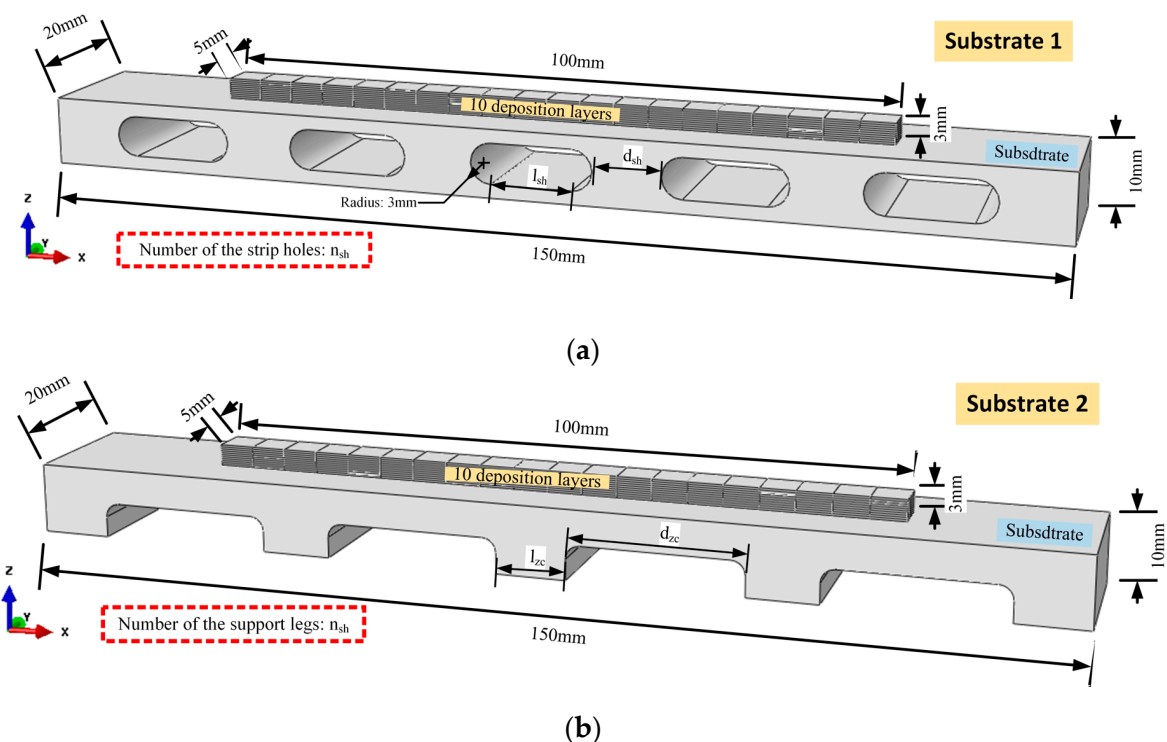

**Figure 5.** Different substrate structure designs used in DED AM: (**a**) substrate with long strip holes; (**b**) substrate with support legs.

## 4. Results and Discussion

### 4.1. Validation of Numerical Model

To validate the accuracy of the sequentially coupled thermomechanical model, the simulated residual distortions of the DED AM product without the substrate design and with the designed long strip holes were compared with the experimentally measured data, which are shown in Figure 6. Although there were issues with the pixel sizes in the experiment and with the element sizes in the numerical model, the interpolated values could be compared with each other. The maximum error between the simulation results and experiment data was less than 5%. The accuracy of the numerical model could be validated.

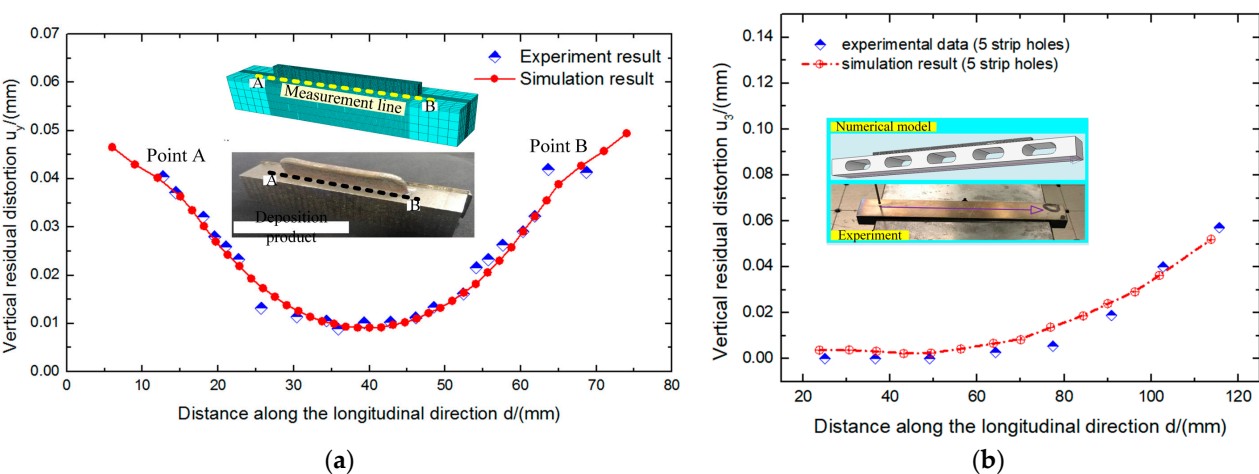

**Figure 6.** Validation of the residual distortions of the DED AM products: (**a**) product without the substrate design; (**b**) product with the substrate design of the long strip holes.

### 4.2. Effect of Constraints on the Residual Distortion and Residual Stress

In the DED AM process, the local high temperature gradient leads to inhomogeneous expansion in the heating process, which is the cause of residual stress and distortion. So, study of the temperature histories and the temperature distribution of the DED AM product is important. Figure 7 shows the temperature distributions of the AM product at different manufacturing times. At the beginning, as shown in Figure 7a, a small melt pool and heat-affected zone could be found in the deposition layer. When the manufacturing time increased, more heat energy accumulated in the layers, so the manufacturing temperature and the size of the melt pool and heat-affected zone all increased, as shown in Figure 7b. When the manufacturing time was 62 s, temperatures of more than 400 °C could be found in the previous deposition layers and the substrate due to the accumulation of heat energy. As shown in Figure 7d, the temperature at the middle of the DED AM product increased to over 800 °C at the manufacturing time of 107 s. The temperature of almost all of the structure in the deposition zone rose to 800 °C when the last layer was deposited. Large expansion distortion of the structure therefore occurred in the heating process from the high manufacturing temperature.

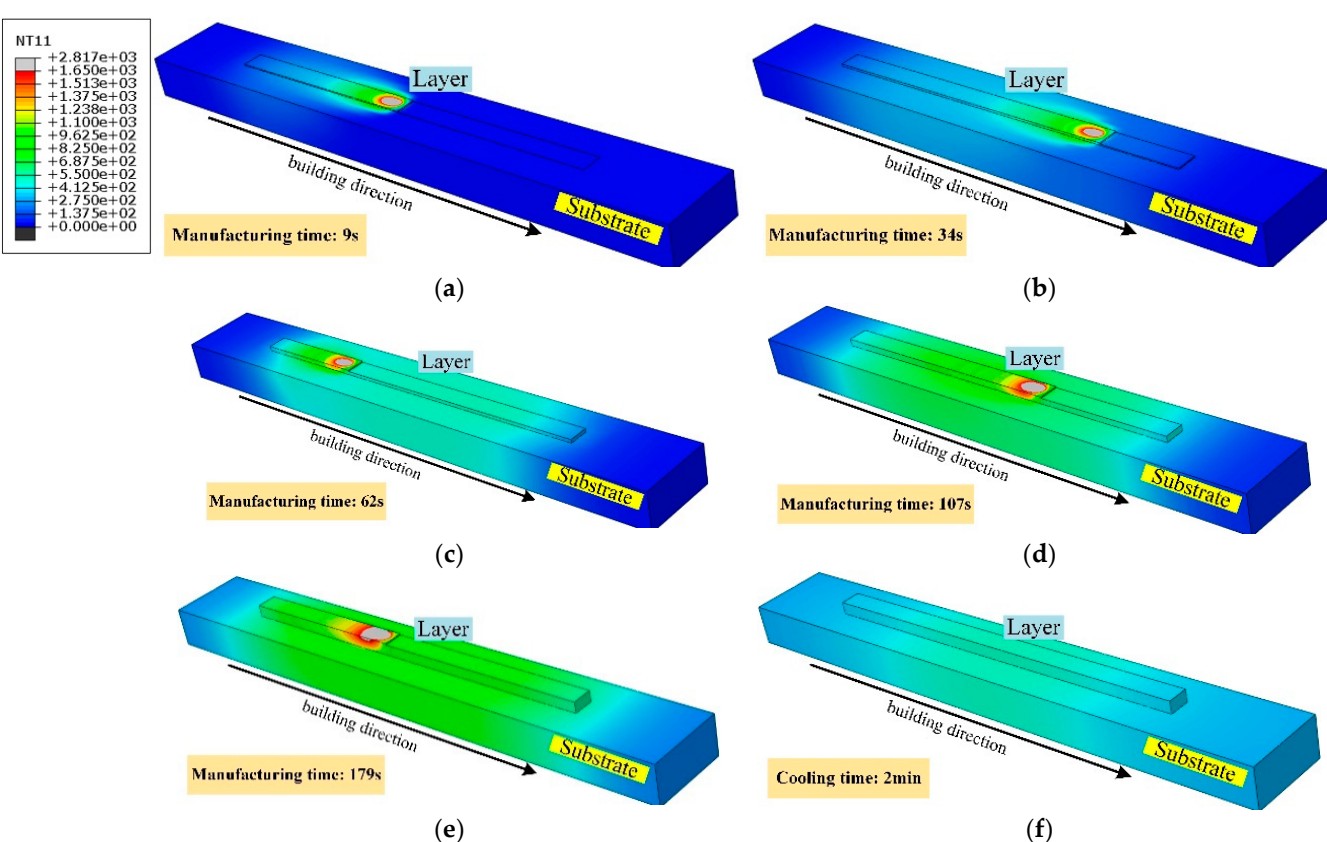

**Figure 7.** Temperature distributions of the thin-walled sample manufactured by DED AM: (**a**) at 9 s; (**b**) at 34 s; (**c**) at 62 s; (**d**) at 107 s; (**e**) at 179 s; (**f**) after cooling for 2 min.

Following the heating process, a 10 min cooling process was applied. The temperature of the DED AM product decreased quickly. At the cooling time of 2 min, the temperature of the deposition product, not only the deposition layers but also the substrate, could decrease to 400 °C. A large amount of shrinking distortion of the structure then took place. The distortion due to expanding and shrinking causes large tensile or compressive thermal strain. This is the reason for the formation of residual distortion and residual stresses in the DED AM product after removing the constraints on the substrate.

In the thermal process, the constraints on the substrate greatly influence the release of the high thermal strain. The effect of the different constraints on the substrate, namely

the constraints on the bottom surface of the substrate (constraints 1) and on one end of the substrate (constraints 2), on the residual distortion and residual stress of the DED AM product were studied, as shown in Figures 8 and 9, respectively. When constraints on the bottom surface of the substrate were used in the DED AM, an upward-bending residual distortion was found in the deposition product after the cooling process, as shown in Figure 8a. The maximum vertical residual distortion was 0.256 mm, which was at the end of the substrate. Vertical residual distortion, about 0.152 mm along the z-axis negative direction, was found in the deposition layers. When the constraints on the substrate were changed to constraint 2, similar upward bending was also found, but with a larger maximum vertical distortion of 0.663 mm at the end of the substrate, as shown in Figure 8b. Meanwhile, the vertical residual distortion along the z-axis negative direction in the deposition layers was greatly decreased. However, most of the layers had a vertical residual distortion along the z-axis positive direction. The weakened constraints on the substrate led to a larger deformation in the z-axis positive direction.

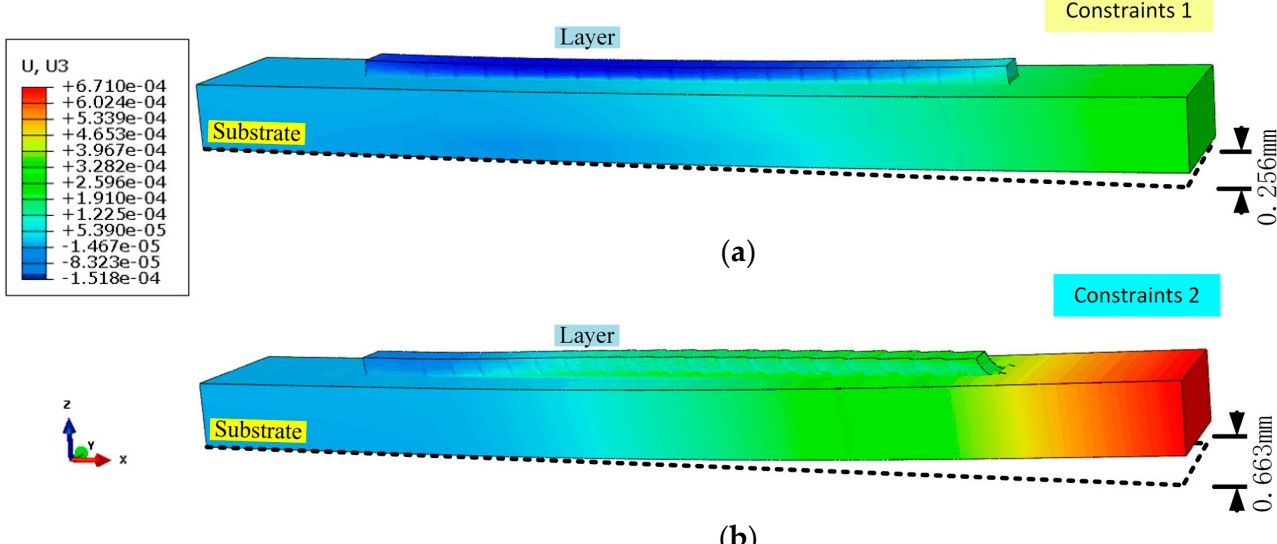

**Figure 8.** Vertical residual distortions of the deposited thin-walled sample: (**a**) constraints on one end of the substrate; (**b**) constraints on the bottom surface of the substrate.

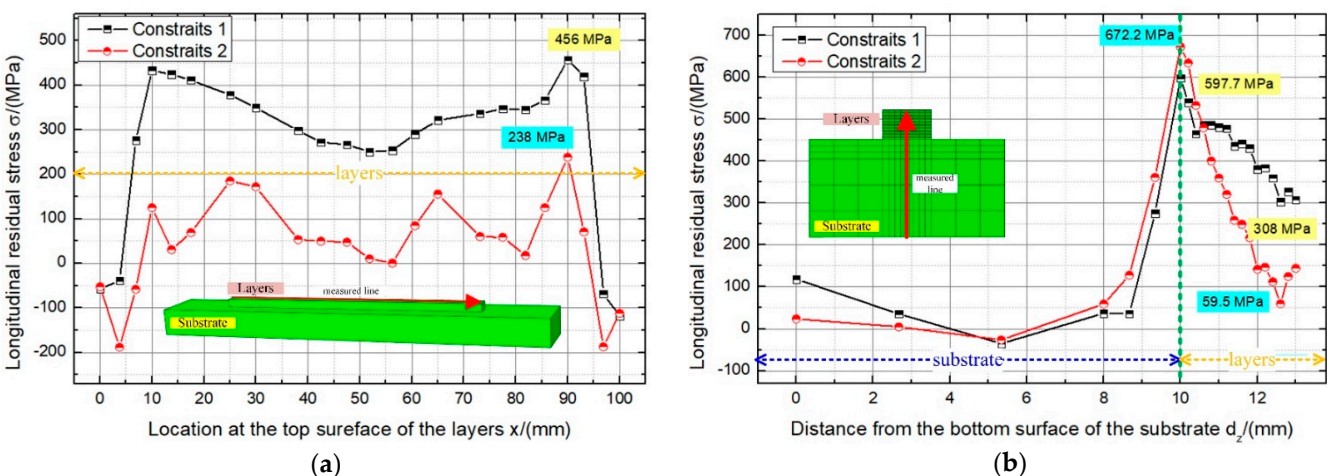

**Figure 9.** Longitudinal residual stresses with different constraints on the substrate: (**a**) on the top surface of the layers; (**b**) in the vertical middle section.

Figure 8 shows the effect of the constraints on the substrate on the residual stresses in the deposition product. As shown in Figure 9a, when the bottom surface of the substrate was constrained, a large tensile residual stress was found on the top surface of the deposition layers. Especially, as the start and end points of the laser movement, in the region near the ends of the layers, the larger tensile residual stresses were found. When the laser was moving in the middle of the layers, the temperature distribution was more homogeneous and less plastic strain was formed. So, the tensile residual stress decreased in locations nearer to the middle section. The maximum tensile residual stress was 456 MPa, which was about half of the yield strength. The minimum tensile residual stress was 250 MPa for a distance of 52 mm and then started increasing.

When constraints on the end of the substrate were used, the tensile residual stress in the layers was greatly decreased, as shown in Figure 9a. The maximum longitudinal tensile stress also decreased to 238 MPa, which was just 28.7% of the yield strength. At the two ends of the layers, the longitudinal tensile residual stress even changed into compressive strength. In the middle of the layers, the longitudinal residual stress decreased by almost 50%. The weaker constraints on the substrate led to the increase in the residual distortion of the deposition product but reduced the residual stress on the top surface of the layers. A similar modeling conclusion was obtained from the longitudinal residual stress results in the vertical middle section, as shown in Figure 9b. In the deposition layers, smaller longitudinal residual stress was found. However, it is noteworthy that the conclusion above was not true in the region near the top surface of the substrate. The largest longitudinal residual stress wase found when the weaker constraints (constraints 2) were used in DED AM. This may be due to the disharmony of the thermal deformation between the deposition layers and the substrate.

The simulated distribution of the residual stress in the deposition layers manufactured by the laser DED AM showed that the large residual stresses occurred in the two sides and the lower layers. The micro cracks and defects in these regions may develop faster with the residual stress, when compared with those in the middle of the layers, especially at the corners of the layers.

### 4.3. Effect of Substrate Structure Design on the Residual Stress

To reduce the longitudinal residual stress in the DED AM product when the constraints on the bottom surface of the substrate were used, the design of the substrate structure was studied in this paper. The longitudinal residual stress on the top surface of the deposition layers and in the middle section of the deposition product are respectively shown in Figures 10 and 11. When long strip holes were designed in the substrate, longitudinal residual stress was also found on the top surface of the deposition layers. The maximum longitudinal residual stress decreased to 328.3 MPa. The tensile residual stress was reduced by 28%, compared with that without the long strip holes. At the end of the layers, the longitudinal compressive residual stress increased to 308 MPa. The residual stress at the start and end points decreased to 39.8% and 37.3% of the yield strength of the Ti-6Al-4V, respectively. When the designed structure was changed to support legs, the smallest longitudinal tensile and compressive residual stresses were found in the layers and at the end of the layers, respectively. Though the maximum longitudinal tensile residual stress was 337.8 MPa, the residual stress in the middle of the layers clearly decreased. Meanwhile, the longitudinal compressive residual stress also decreased to 243.7 MPa. The structure designed with the support legs was better than that with the long strip holes.

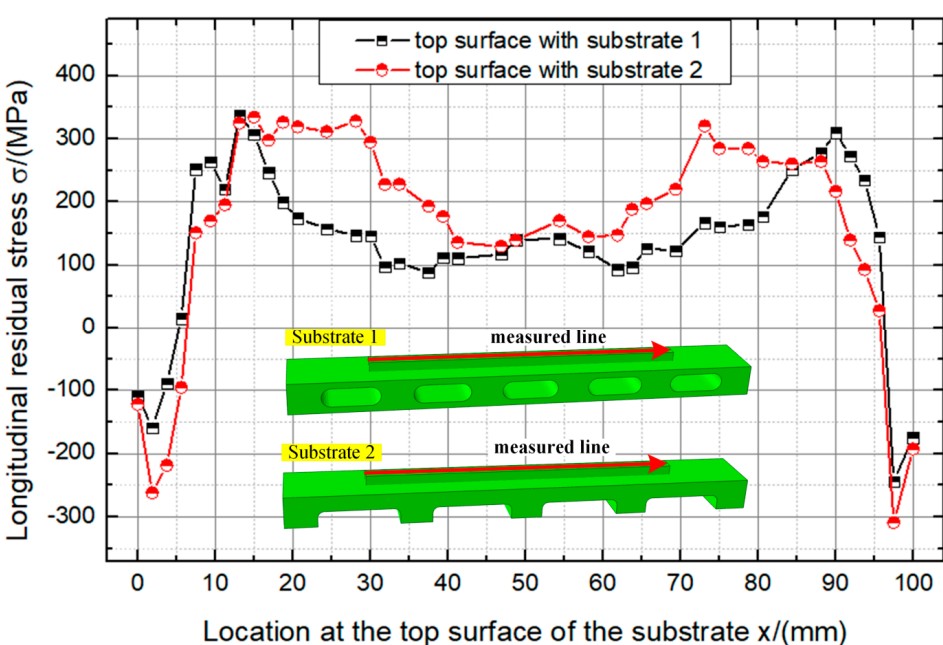

**Figure 10.** Longitudinal residual stresses on the top surfaces of the deposition layers with different substrate structures.

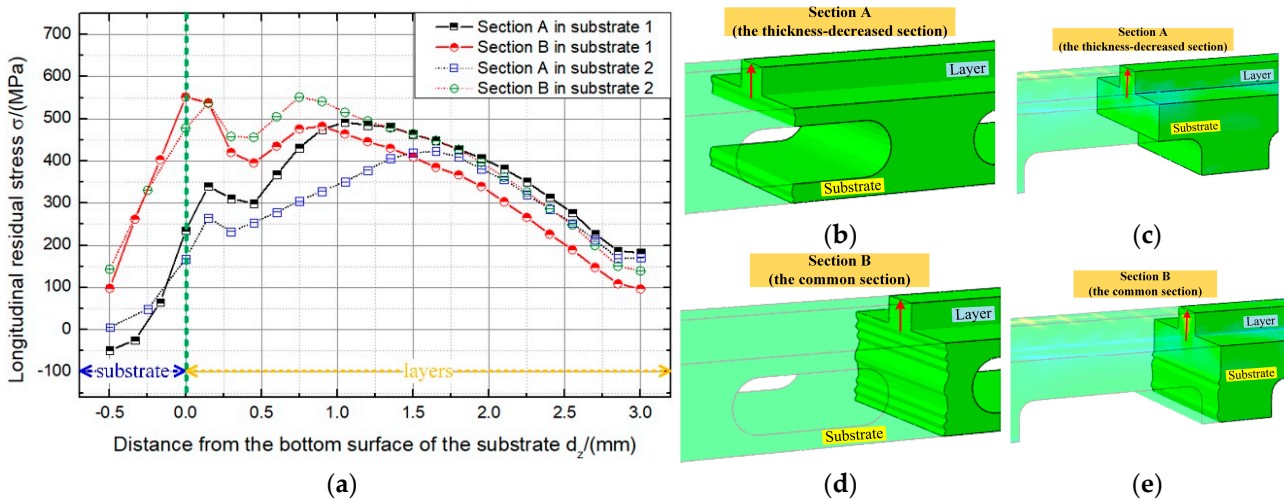

**Figure 11.** Longitudinal residual stresses with different substrate structures: (**a**) distribution of residual stress in vertical sections; (**b**) section A in substrate 1; (**c**) section A in substrate 2; (**d**) section B in substrate 1; (**e**) section B in substrate 2.

When the longitudinal residual stress in the middle section of the deposited product was studied, a similar conclusion was obtained. The measured lines in different sections are respectively shown in Figure 11b–e. As shown in Figure 11, the longitudinal residual stresses varied with the location of the measured section. In the first several layers, the tensile residual stress in the common section was larger than that in the reduced-thickness section; but it was smaller in the upper layers. This means that, with the weakened stiffness near substrate 1, the accumulated thermal strain was transferred to the upper layers. When the support legs were designed in the substrate, the stiffness of the lower layers increased compared with that in substrate 1, so the longitudinal tensile residual stress increased slightly in the layers, as shown in Figure 11a. However, when the measured section was moved to the reduced-thickness section, there was a clear decrease in the longitudinal tensile residual stress. The maximum tensile residual stress decreased to 423.2 MPa, while

it was 491.6 MPa in substrate 1. Based on the discussion above, it could be construed that the structure designed with the support legs was more beneficial for manufacturing DED AM products with less longitudinal residual stress.

### 4.4. Study on the Designed Parameter of the Support Legs

After choosing the support legs as the method for controlling the residual stress in DED AM products, the effects of the design parameters on the residual stress need to be discussed further. The von Mises stresses in DED AM products with different substrate support legs are shown in Figure 12. With the increase in the number of support legs, larger von Mises stresses were found in the deposition layers, as shown in Figure 12a–c. When four support legs were used in the design, the von Mises stresses in the middle of the layers were larger than those at the two ends of the layers. When the number of support legs was increased, the von Mises stresses in the middle decreased but increased at the two ends of the layers. From the comparison of the von Mises stresses of the middle section with different designs of support legs, it could be seen that the larger the distance between the two support legs, the larger the longitudinal tensile residual stress.

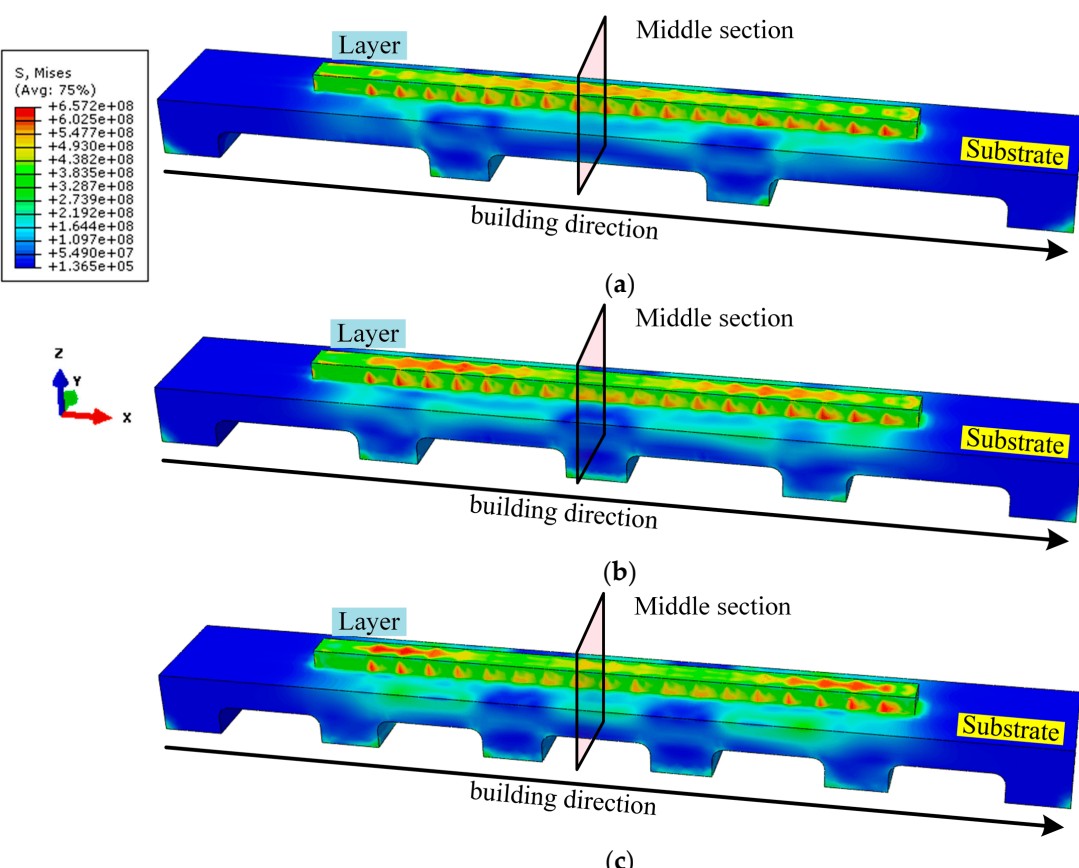

**Figure 12.** Residual von Mises stresses with different numbers of support legs: (**a**) 4 legs; (**b**) 5 legs; (**c**) 6 legs.

Figures 13 and 14 show in detail the residual stresses on the top surface of the layers and in the middle section of the deposition product. With the increase of the number of support legs, the longitudinal tensile residual stress on the top surface of the layers increased and the compressive stress decreased, as shown in Figure 13. When four support legs were used, the maximum tensile residual stress was 252.6 MPa. When five support legs were used, the maximum increased to 344.4 MPa. When the number of support legs increased to six, the maximum increased by 79.5% with a value of 453.5 MPa. Increasing the number of support legs increased the stiffness of the substrate. As a result, the thermal

deformation of the deposition layers was more strongly constrained. The greater the accumulation of thermal strain, the larger the longitudinal tensile residual stress on the top surface of the deposition layers. The phenomenon of the residual stress increasing with the increase of the number of support legs also occurred in the comparison of the longitudinal residual stress in the reduced-thickness section of the deposited product, as shown in Figure 14. When the number of support legs was increased from four to six, the maximum longitudinal residual stress in the deposition layers increased from 371.8 MPa to 503.5 MPa. It increased by 8.1% and 35.4%, respectively, when the support legs were increased to five and six.

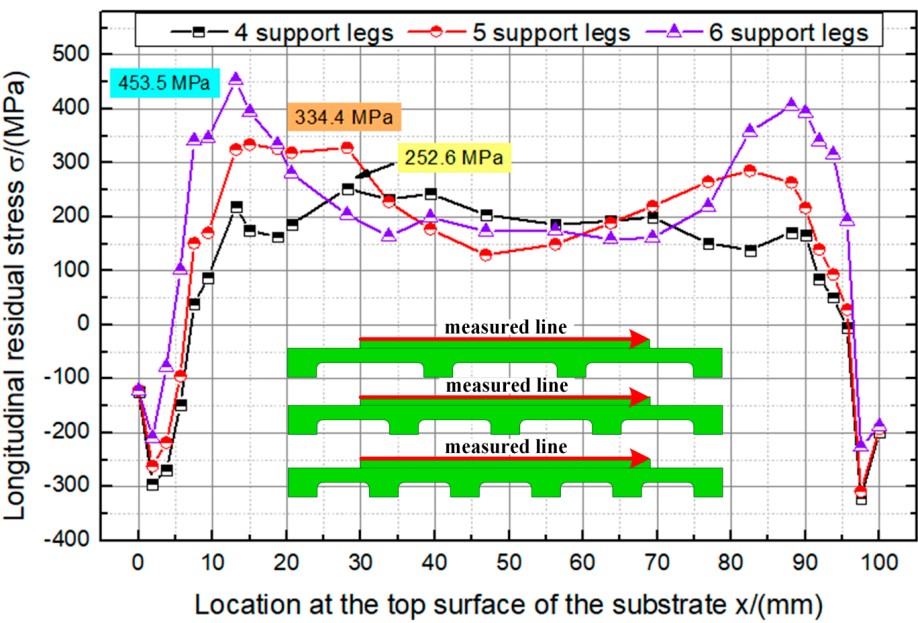

**Figure 13.** Longitudinal residual stresses on the top surfaces of the deposition layers with different numbers of support legs.

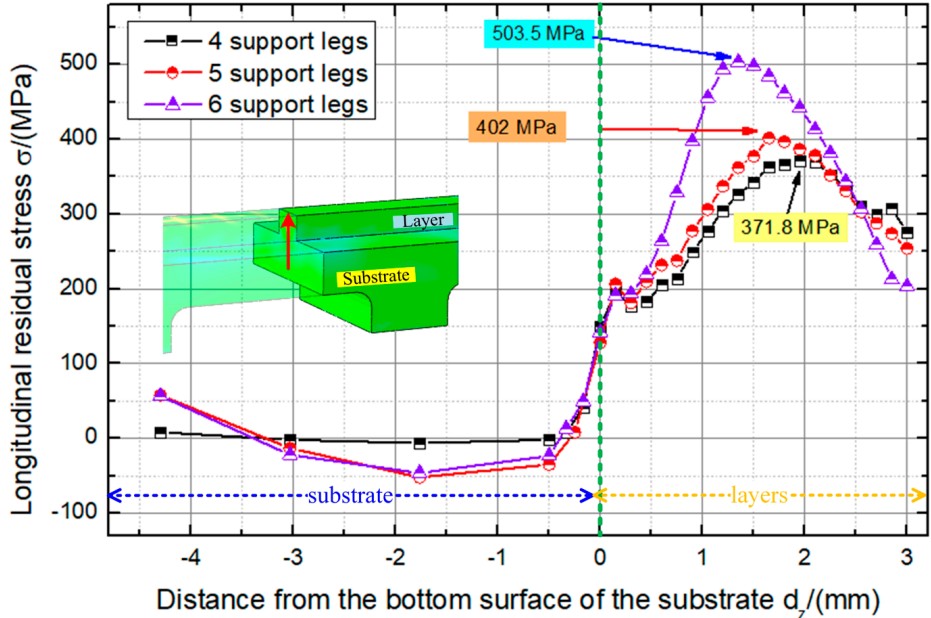

**Figure 14.** Longitudinal residual stresses of the reduced-thickness section with different numbers of support legs.

## 5. Conclusions

The effects of the constraints and substrate structure on the residual stresses in DED AM products were studied in this paper using a sequentially coupled thermomechanical model. The accuracy of the numerical model was also validated by comparing the simulation results of the residual distortion of the DED AM products with the experimental data. The main conclusions are listed below:

1. A new structure design of the substrate used in laser DED AM was proposed to control the residual stress.
2. Compared with the application of constraints on the bottom surface of the substrate, less residual stress can be found in the deposition product when the constrains were applied on one end of the substrate.
3. By designing a local reduced-thickness region in the substrate, the residual stress in the DED AM product could be decreased more than 30%. The fewer support legs, the smaller the residual stress.

Based on the research in the current work, a new controlling method by a topology design of the substrate could be used to better relax the plastic strain and the thermal deformation. As a result, laser DED AM products with decreased residual stress can be manufactured.

**Author Contributions:** Data curation, H.J. and P.G.; Investigation, H.J., P.G., J.-Q.C., and W.-W.L.; Project administration, Z.Z.; Resources, Z.-M.L.; Supervision, Z.-M.L.; Validation, P.G., Z.Z., and W.-W.L.; Writing–original draft, H.J.; Writing–review & editing, P.G. and Z.Z. All authors have read and agreed to the published version of the manuscript.

**Funding:** This research received no external funding.

**Institutional Review Board Statement:** Not applicable.

**Informed Consent Statement:** Not applicable.

**Data Availability Statement:** Data available on request.

**Acknowledgments:** This study was supported by the Liaoning Provincial Natural Science Foundation [2019-KF-05-07, Zhao Zhang], NSFC Henan joint fund, National Natural Science Foundation of China [U1804254, Zhong-Ming Liu], and National Natural Science Foundation of China [52175455, Wei-wei Liu].

**Conflicts of Interest:** The authors declare that they have no conflicts of interest.

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
