# Peer review of "Numerical Studies of the Effects of the Substrate Structure on the Residual Stress in Laser Directed Energy Additive Manufacturing of Thin-Walled Products"

_metals, doi:10.3390/met12030462_

Round 1
Reviewer 1 Report
Please explain the constraints 1 and 2 better.

Reviewer 2 Report
Please see the attachment.

Reviewer 3 Report
In this paper the authors used a sequentially coupled thermo-mechanical model to simulate the residual stresses in DED AM products with different substrate structures and constraints.
In Thermal Analysis, the powder that reaches the substrate has its material property, which are quite different to the solid material, the authors have not considered that aspect of the problem.
The double ellipsoid heat source is used, in general, for arc source, why the authors have used for modelling the laser source?
How is the numerical model validated exactly?
What do the authors estimated for an impact of their research?
References from the authors are numerous and not always justified. For example, the way of introduction of the thermal model looks like a pretext for refs 32-35 from the authors. I suggest to the authors to carefully evaluate which of their papers must be correctly cited in the references.
Round 2
Reviewer 2 Report
Accept.